# Prospective study of gut hormone and metabolic changes after laparoscopic sleeve gastrectomy and Roux-en-Y gastric bypass

Rachel Arakawa[1], Gerardo Febres[1], Bin Cheng[2], Abraham Krikhely[3], Marc Bessler[3], Judith Korner[1¤]*

1 Division of Endocrinology & Metabolism, Department of Medicine, College of Physicians and Surgeons, Columbia University, New York, New York, United States of America, 2 Department of Biostatistics, College of Physicians and Surgeons, Columbia University, New York, New York, United States of America, 3 Division of Minimal Access/Bariatric Surgery, Department of Surgery, College of Physicians and Surgeons, Columbia University, New York, New York, United States of America

¤ Current address: Division of Endocrinology & Metabolism, College of Physicians and Surgeons, Columbia University, New York, New York, United States of America

* jk181@cumc.columbia.edu

**Data Availability Statement:** All relevant data are within the manuscript and its Supporting Information files.

## Abstract

### Background

Laparoscopic sleeve gastrectomy (SG) has surpassed Roux-en-Y gastric bypass (RYGB) as the most prevalent bariatric procedure worldwide. While RYGB and SG demonstrate equivalent short-term weight loss, long-term weight loss tends to be greater after RYGB. Differences in the effect of these procedures on gastrointestinal hormones that regulate energy homeostasis are felt to partially underlie differences in outcomes. The objective of this study was to prospectively quantify blood levels of gut hormones of energy and glucose homeostasis at one year follow up to delineate possible reasons for greater efficacy of RYGB over SG in achieving weight loss.

### Methods

Patients undergoing SG (n = 19) and RYGB (n = 40) were studied before surgery and at 2, 12, 26, and 52 weeks postoperatively. Blood samples drawn in the fasted state and after a liquid mixed meal were assayed at baseline, 26, and 52 weeks for peptide YY (PYY), glucagon-like peptide-1 (GLP-1), ghrelin, insulin, glucose, and leptin. Fasting and postprandial appetitive sensations were assessed by visual analog scale.

### Results

At 1 year there was greater weight loss in RYGB compared with SG patients (30% vs 27%; $P = 0.03$). Area under the curve (AUC) after the mixed meal for PYY was greater in RYGB patients ($P<0.001$). RYGB patients had significant increases in GLP-1 AUC compared to baseline ($P = 0.002$). Ghrelin levels decreased only after SG compared to baseline ($P<0.001$) but were not significantly different from RYGB. There was a trend toward decreased sweet cravings in RYGB patients ($P = 0.056$).

**Funding:** This work was supported by the National Institutes of Health [DK072011 to JK, T32 DK07271 to Dr. Domenico Accili, UL1TR001873 and UL1TR000040 to Dr. Muredach Reilly]. RA was supported in part by the National Institutes of Health (T32 DK 007559-29). The funders had no role in study design, data collection and analysis, decision to publish, or preparation of the manuscript.

**Competing interests:** The authors have read the journal's policy and have the following competing interests: JK participates on advisory boards for Digma Medical, GI Dynamics, and Esquagama. She receives financial compensation and stock options from GI Dynamics and Esqagama. She receives stock options from Digma Medical. She does not receive financial support via salary for her participation on these advisory boards. MB is a founder of Endobetes and has a patent application titled "Lumen Reinforcement and Anchoring System" application no. 16/046592. The patent is focused on a method to anchor devices in the GI tract and is assigned to Endobetes, a medical device startup with a goal of developing endoscopic devices for treatment of obesity and diabetes. MB owns stock in Endobetes and has an equity position, but does not receive financial support via salary as the founder of Endobetes. AK is a consultant for Intuitive Surgical, CLG, and CSATs. He was paid by Intuitive surgical to give several talks and serve as an OR surgery instructor/proctor. CLG is a third party consulting firm that connects firms with experts and AK was paid to offer an opinion on topics related to robotic surgery. He was paid by CSATs to review videos of surgeries and provide critique to the surgeons. He does not receive any financial support via salary or research funding from these firms. This does not alter our adherence to PLOS ONE policies on sharing data and materials.

## Conclusions

Differences in gastrointestinal hormones that regulate energy and glucose homeostasis are a possible mechanism for greater efficacy of RYGB compared to SG.

## Introduction

Obesity is a major health problem with increasing prevalence and is associated with increased mortality [1–3]. Bariatric surgery produces greater weight loss, improvement in comorbidities, and survival compared to non-surgical treatment for obesity [4–6]. There has been a 20-fold increase in procedures performed annually over the past 20 years in the United States. In the past decade, there has been a rapid shift in procedure type performed. Roux-en-Y gastric bypass (RYGB) predominated until the late-2000s, then was surpassed by sleeve gastrectomy (SG), now the most commonly performed procedure in the US and worldwide [7, 8]. While prior trials were not powered to compare groups [9, 10], long-term data comparing the efficacy of the surgeries has emerged favoring RYGB. A 65,000-patient retrospective multicenter cohort demonstrated 6.7% greater total weight loss after RYGB compared to SG at 5 year follow up [11]. In a longitudinal study, mean weight loss at 7 years follow up was significantly higher in RYGB (30.4%) vs SG (23.6%) [12]. A meta-analysis of over 5000 patients showed superior weight loss after RYGB with a mean difference of 17% at greater than 5 years follow up [13]. Additionally, RYGB has been shown to be superior to SG in achievement of type 2 diabetes remission. In a single center triple-blind randomized trial, diabetes remission was achieved in 75% of RYGB vs. 48% of SG subjects at 1 year [14]. In a 9700-patient retrospective multicenter cohort the diabetes remission rate was approximately 10% higher in patients who had RYGB compared to SG and they experienced significantly lower relapse rates at 5 years after surgery [15]. Several prior trials also showed more favorable remission rates after RYGB compared to SG [4, 9, 10].

Changes in gut hormones of energy and glucose homeostasis following these procedures are felt to partially underlie differences in outcomes [16–19]. A variety of hormones secreted from the gastrointestinal tract communicate with peripheral tissues and the central nervous system to regulate glucose homeostasis and energy balance. The gut hormones of interest in this report are peptide YY (PYY), glucagon-like peptide-1 (GLP-1), and ghrelin. PYY and GLP-1 are secreted by distal intestinal L cells to decrease appetite, increase satiety, and slow gut motility. [20]. Additionally, PYY improves insulin sensitivity [21, 22] and GLP-1 functions as an incretin to potentiate glucose-stimulated insulin release. Ghrelin is an appetite stimulating neuropeptide secreted by the gastric fundus and proximal small intestine to increase food intake, gut motility, and decrease insulin secretion [23, 24]. Ghrelin levels increase during fasting, are suppressed following meal intake, and increase in response to diet-induced weight loss or adjustable gastric banding [25–27].

The objective of this study was to quantify blood levels of GI tract hormones at one year follow up to delineate possible reasons for greater efficacy of RYGB over SG in achieving weight loss and metabolic improvements.

## Materials and methods

### Protocol

We certify that all applicable institution and government regulations concerning the ethical use of human volunteers was obeyed during this study. The study was approved by the

Columbia University Institutional Review Board and written informed consent was obtained from all subjects. Patients were recruited if they were above the age of 21, scheduled to undergo either RYGB or SG, and did not use weight loss medications within 90 days prior to enrollment. This cohort consists of 59 subjects, recruited from April 2003 to September 2017, who underwent either RYGB (n = 40) or SG (n = 19) and had 1 year of follow up data. Sample size was based on prior and ongoing work comparing RYGB and laparoscopic adjustable gastric banding (LAGB) in which differences in gut hormone levels were observed in cross-sectional and prospective studies [27–29].

The choice of bariatric procedure was based on patient and surgeon preference. RYGB entailed creation of a 15-30ml gastric pouch (divided from the proximal lesser curvature of the stomach and excluded the fundus) and anastomosed to a Roux limb of jejunum created by division of the jejunum 50-100cm distal to the ligament of Treitz and anastomosing the afferent biliopancreatic limb of the jejunum 100-150cm distally. Division of the vagus nerve and its branches was avoided. SG was performed with a 40Fr bougie as a template aligned along the lesser curvature. Gastric transection was performed beginning 5cm proximal to the pylorus and extended to the Angle of His.

Participants were seen at baseline, 2, 12, 26, and 52 weeks for measurement of body weight and waist circumference. At baseline, week 26, and week 52 blood was drawn in a fasted state followed by consumption of a liquid test meal over a 15 minute period (Optifast, Novartis, Minneapolis, MN, USA; 474 ml, 320 kcal, 50% carbohydrate, 35% protein, 15% fat). Venous blood was then drawn at 15, 30, 60, 90, and 120 minutes from the end of the meal. After centrifugation at 4ºC, both serum and plasma were stored at -80ºC. Subjects completed a validated visual analog scale (VAS) questionnaire [30, 31] in the fasted state and at 30, 60, 90, and 120 minutes after the test-meal. The VAS consisted of 100-mm lines with words anchored at each end describing extreme sensations of hunger, satiety, sweet cravings, and nausea or abdominal comfort. Subjects were asked to make a vertical mark across the line corresponding to their feelings. Quantification was performed by measuring the distance from the left end of the line to the mark.

## Hormone assays

Total PYY was measured by ELISA (Millipore, MO, USA) with a sensitivity of 10pg/ml. Total GLP-1 was measured by RIA (Millipore, MO, USA) after alcohol extraction according to manufacturer's protocol with a sensitivity of 3pM and recovery in each assay tested by parallel extraction of standards. Leptin was measured with a human RIA kit (LINCO Research, Inc, St. Charles, MO) using a $^{125}$I-iodinated human leptin tracer. Total plasma immunoreactive ghrelin was measured by a RIA kit (Phoenix Pharmaceuticals, Belmont, CA) using $^{125}$I-iodinated ghrelin tracer and a rabbit polyclonal antibody against full-length, octanolyated human ghrelin that recognizes the acyl and des-acyl forms of the hormone, with the lower limit of detection of 20 pg/ml. Plasma glucose was measured by the hexokinase method. Plasma insulin was measured with the Immulite Analyzer (Diagnostic Products Corp., Los Angeles, CA) with the lower limit of detection of 2ulU/ml. An aliquot from a pool of plasma was included in each assay to ensure that there was no change over time. All samples were assayed in duplicate.

## Statistical analysis

Statistical model estimated means and standard errors are presented.

Linear mixed effects models were used to analyze the outcome variables. Specifically, the main predictors were procedure (RYGB versus SG), time (in week, treated as categorical for possible exploration of nonlinear temporal trend), and their interactions. Random intercept effects were included in the model to account for between-subject variation and within-subject

correlation, which was equivalent to a compound symmetric covariance structure. No other covariance structures were explored due to limited sample size. The estimated mean and standard error at each time point was reported from the model fit and compared using a Wald test. A significant interaction effect was considered as a confirmation of procedure effect. The linear mixed model was chosen to assess the procedure effect for multiple reasons. First, compared with generalized estimating equation, another main approach to modeling longitudinal data, the mixed model is a conditional model and hence is able to capture the subject specific features, which matches with the purpose of clinical trials. Second, the mixed model is a likelihood-based model, and hence would remain appropriate even if there are missing data. In fact, it is theoretically shown that using all available data (i.e., even those subjects with partial data) the mixed model would yield correct parameter estimations under the missing at random (MAR) assumption, one which was deemed reasonable in practice and its violation could not be confirmed in such a small sample size. A $P < .05$ was considered statistically significant. SAS version 9.4 was used in the analysis. Total area under the curve (AUC) was calculated using the trapezoidal rule from 0–120 min, with the exception of GLP-1, which was calculated from 0–60 min. Insulin resistance was calculated using the homeostatic model of assessment (HOMA-IR) [32]. The Pearson correlation analysis was used for HOMA-IR and percentage weight loss.

# Results

## Study subjects

There were 59 subjects enrolled consisting of 63% Hispanic, 37% non-Hispanic, 22% African American, and 78% Caucasian adults; 40 subjects underwent RYGB (8M/32F) and 19 underwent SG (4M/15F).

Baseline characteristics were similar for age (RYGB 43.6± 1.6 years; SG 44.5 ± 1.8 years $P = 0.34$) and BMI (RYGB 46.4 ± 5.5 kg/m$^2$; SG 45.6 ± 8 kg/m$^2$ $P = 0.32$). Baseline characteristics and anthropometric data are presented in Table 1. While weight was similar at baseline

**Table 1. Baseline characteristics of participants.**

|  | Sleeve Gastrectomy (n = 19) | Roux-en-Y Gastric Bypass (n = 40) |
|---|---|---|
| **Age--yr** |  |  |
| Mean | 44.5 ± 1.8 | 43.6 ± 1.6 |
| Range | 27–57 | 22–64 |
| **Sex** |  |  |
| Female—no. (%) | 15 (78.9) | 32 (80.0) |
| Male—no. (%) | 4 (21.1) | 8 (20.0) |
| **Race** |  |  |
| White—no. (%) | 12 (63.2) | 34 (85.0) |
| Black—no. (%) | 7 (36.8) | 6 (15.0) |
| **Ethnicity** |  |  |
| Hispanic—no. (%) | 12 (63.2) | 25 (62.5) |
| Non-Hispanic—no. (%) | 7 (36.8) | 15 (37.5) |
| **Weight/BMI** |  |  |
| Weight (kg) | 124.0 ± 4.4 | 123.6 ± 3.0 |
| Waist (cm) | 124.4 ± 3.0 | 125.2 ± 2.0 |
| BMI (kg/ m$^2$) | 45.6 ± 1.4 | 46.4 ± 0.9 |

Results are presented as mean ± SEM unless indicated otherwise.

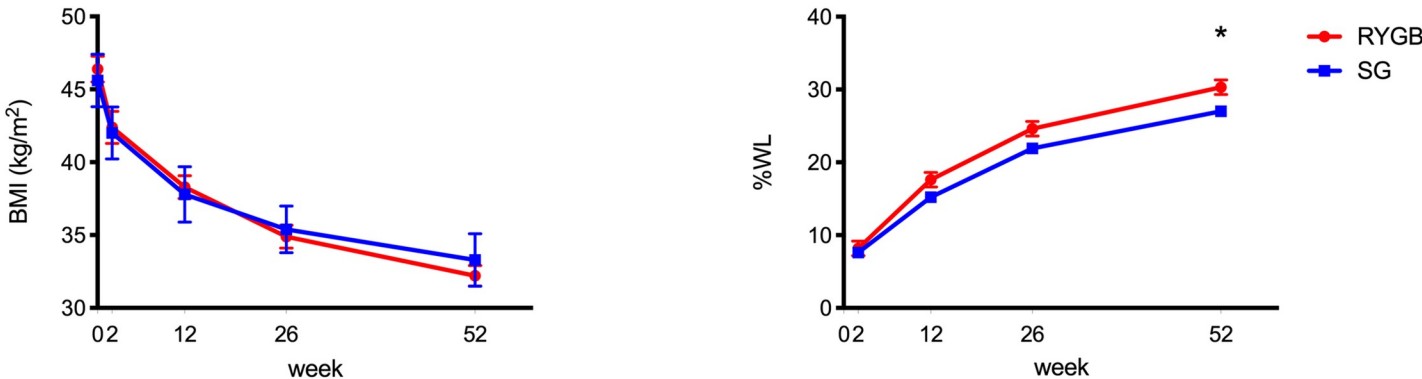

**Fig 1. BMI and percentage weight loss (%WL) after SG and RYGB.** Values are reported as mean ± SEM. *$P$ <0.05 between group difference.

(RYGB 123.6 ± 19.2 kg; SG 124 ± 26.9 kg $P$ = 0.47), there was a significantly higher percentage weight loss after RYGB compared to SG (30.3% vs 27.0%, $P$ = 0.03) at 52 weeks (Fig 1).

## Hormone and glucose levels

Values of blood hormone levels are presented in Table 2 and Fig 2. Fasting levels of PYY and GLP-1 did not change after RYGB or SG. Postprandial PYY increased significantly after RYGB at 52 weeks, with AUC levels approximately two-fold greater than baseline. Postprandial PYY increased to a lesser degree after SG and AUC levels were significantly lower compared with

**Table 2. Baseline characteristics and changes over time in body weight, glucose and plasma hormone levels after SG and RYGB.**

| | Sleeve Gastrectomy | | | Roux-en-Y Gastric Bypass | | |
|---|---|---|---|---|---|---|
| | Week 0 | Week 26 | Week 52 | Week 0 | Week 26 | Week 52 |
| **Wt/BMI** | | | | | | |
| Wt (kg) | 124.0 ± 4.4 | 96.8 ± 4.4*** | 90.4 ± 4.4*** | 123.6 ± 3.0 | 92.9 ± 3.1*** | 85.8 ± 3.1*** |
| Waist (cm) | 124.4± 3.0 | 102.1 ± 3.0*** | 97.5 ± 3.0*** | 125.2 ± 2.0 | 104.9 ± 2.0*** | 98.3 ± 2.0*** |
| BMI (kg/m²) | 45.6 ± 1.4 | 35.4 ± 1.4*** | 33.3 ± 1.4*** | 46.4 ± 0.9 | 34.9 ± 0.9*** | 32.3 ± 0.9*** |
| WL (%) | n/a | 21.9 ± 1.3*** | 27.0 ± 1.3*** | n/a | 24.6 ± 1.0*** | 30.3 ± 1.0***O |
| **Glucose(mg/dl)** | | | | | | |
| Fasting | 102.4 ± 5.4 | 81.7 ± 5.4*** | 81.0 ± 5.6*** | 112.3 ± 3.7 | 91.8 ± 3.8*** | 86.3 ± 3.7*** |
| AUC x 10³ | 14.2 ± 0.9 | 10.3 ± 0.9*** | 9.6 ± 1.0*** | 15.0 ± 0.7 | 11.3 ± 0.7*** | 10.6 ± 0.7*** |
| **Insulin (uIU/ml)** | | | | | | |
| Fasting | 17.0 ± 4.1 | 12.6 ± 4.1 | 8.5 ± 4.4 | 20.7 ± 2.8 | 14.7 ± 3.0 | 7.0 ± 2.8*** |
| AUC x 10³ | 8.3 ± 0.8 | 5.5 ± 0.8*** | 5.2 ± 0.9*** | 6.8 ± 0.6 | 4.5 ± 0.6*** | 4.0 ± 0.6*** |
| **HOMA-IR** | - | - | - | - | - | - |
| | | 4.5 ± 1.1 | 2.5 ± 1.1 | 1.7 ± 1.3* | 6.0 ± 0.8 | 3.2 ± 0.7** | 1.5 ± 0.8*** |
| **PYY (pg/ml)** | | | | | | |
| Fasting | 81.1 ± 21.5 | 75.8 ± 21.5 | 75.0 ± 22.6 | 92.6 ± 14.9 | 123.5 ± 15.5* | 119.7 ± 14.9 |
| AUC x 10³ | 13.1 ± 4.6 | 24.4 ± 4.6* | 18.8 ± 4.9 | 16.0 ± 3.5 | 37.9 ± 3.4***O | 39.1 ±3.2***OOO |
| **GLP-1 (pmol/l)** | | | | | | |
| Fasting | 19.1 ± 3.1 | 18.1 ± 3.2 | 16.5 ± 3.1 | 14.5 ± 2.3 | 13.5 ± 2.3 | 13.1 ± 2.2 |
| AUC x 10³ | 1.6 ± 0.2 | 2.4 ± 0.2** | 1.8 ± 0.2 | 1.4 ± 0.2 | 2.4 ± 0.2*** | 2.2 ± 0.2** |
| **Ghrelin (pg/ml)** | | | | | | |
| Fasting | 518.6 ± 35.2 | 230.8 ± 36.7*** | 224.9 ± 43.5*** | 347.1 ±24.2OOO | 347.4 ± 25.0 OO | 347.1 ± 24.7 O |
| AUC x 10³ | 55.0 ± 3.7 | 25.2 ± 3.9*** | 25.0 ± 4.6*** | 36.3 ± 2.7OOO | 35.3 ± 2.7O | 33.7 ± 2.6 |

*(Continued)*

**Table 2.** (Continued)

| | Sleeve Gastrectomy | | | Roux-en-Y Gastric Bypass | | |
|---|---|---|---|---|---|---|
| | Week 0 | Week 26 | Week 52 | Week 0 | Week 26 | Week 52 |
| **Leptin (ng/ml)** | | | | | | |
| Fasting | 56.7 ± 3.7 | 27.0 ± 3.7*** | 26.0 ± 3.9*** | 43.2 ± 2.5 ᴼᴼ | 17.2 ± 2.6***ᴼ | 15.6 ± 2.5*** ᴼ |

Values presented are linear mixed model mean ± SEM. P value compared with Week 0 within-group

*P < 0.05

**P < 0.01

***P < 0.001. P value compared with SG values at same time point: P < 0.05

ᴼᴼP < 0.01

ᴼᴼᴼP < 0.001. HOMA-IR units: (mmol x μIU x l⁻²). AUC x 10³ are integrated over 0–120 min with the exception of AUC for GLP-1 which was determined from 0–60 min.

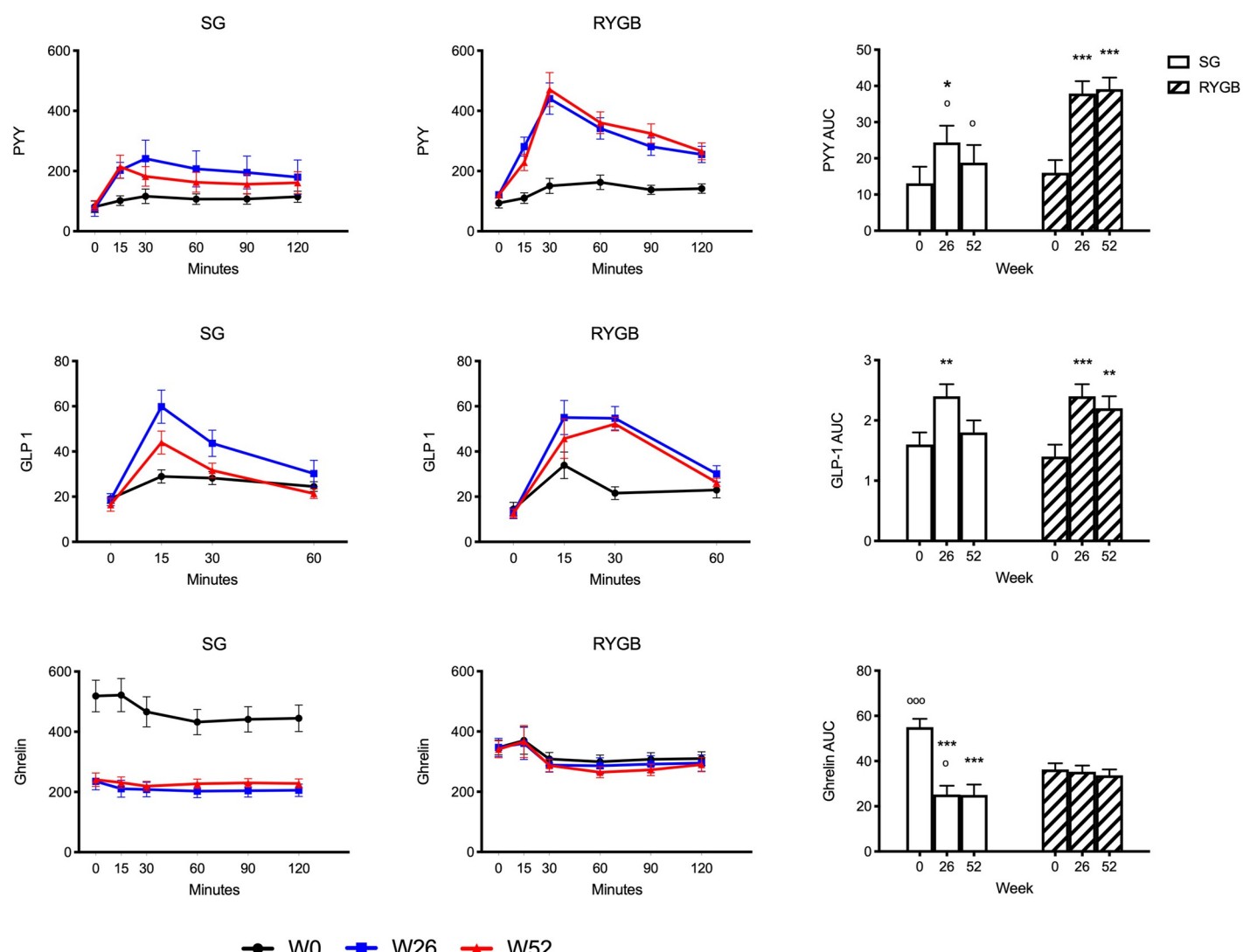

**Fig 2. Fasting and postprandial gut hormone levels after SG and RYGB.** Values are reported as model estimated mean ± SEM. *P <0.05, **P <0.01, ***P <0.001 within group difference; ᵒP <0.05, ᵒᵒP <0.01, ᵒᵒᵒP <0.001 between group difference.

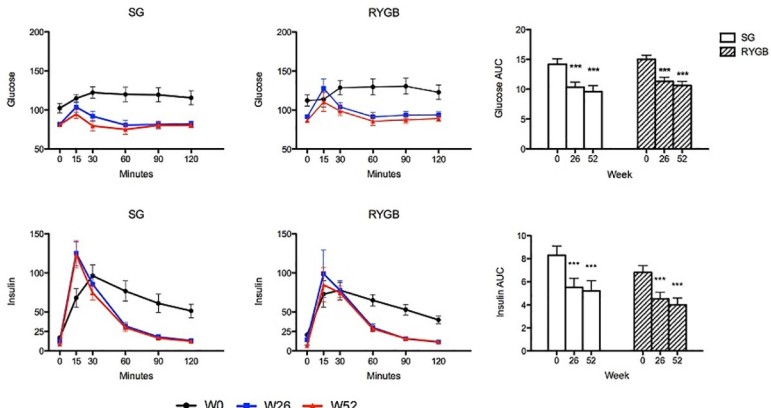

**Fig 3. Fasting and postprandial glucose and insulin.** Values are reported as model estimated mean ± SEM. $^*P < 0.05$, $^{**}P < 0.01$, $^{***}P < 0.001$ within group difference.

RYGB. While PYY AUC increased after SG compared to baseline at 26 weeks, significance was not maintained at 52 weeks. GLP-1 levels increased in both groups. Similar to PYY, GLP-1 AUC following SG was significantly higher than baseline at 26 weeks, but was not maintained at 52 weeks. GLP-1 AUC increased significantly after RYGB from baseline at 26 weeks and persisted at 52 weeks. At baseline, ghrelin levels were greater in the SG group compared with RYGB. However, ghrelin levels drastically decreased after SG such that levels became significantly lower post-surgery compared with baseline and compared with the same time-points post-RYGB. In contrast, there was no significant change in ghrelin levels from baseline after RYGB. Data were further analyzed as percent change in AUC from baseline: at 26 weeks percent change was -46.6% for SG and 11.3% for RYGB (P <0.01); and at 52 weeks percent change was -45.2% for SG and 13.7% for RYGB (P <0.01). Leptin levels decreased significantly in both groups and were consistently lower at baseline and post-surgery in the RYGB group.

Both groups had a similar percentage of subjects with type 2 diabetes mellitus (21% SG, 27% RYGB, *P* = 0.59). Combined data from subjects with and without diabetes are presented (Table 2). Fasting and postprandial glucose decreased in both groups without significant differences between procedures. Insulin also decreased, with both groups demonstrating early peak levels at 15 minutes post-test meal followed by a rapid decline such that insulin AUC decreased significantly in each (Table 2 and Fig 3). HOMA-IR decreased in both groups. While there was a significant negative correlation between HOMA-IR and percent weight loss in SG (r = -0.65, P = 0.008) there was no correlation in RYGB (r = -0.15, P = 0.4) (Fig 4). Data were similar when subjects with diabetes were excluded from the correlation analysis; there

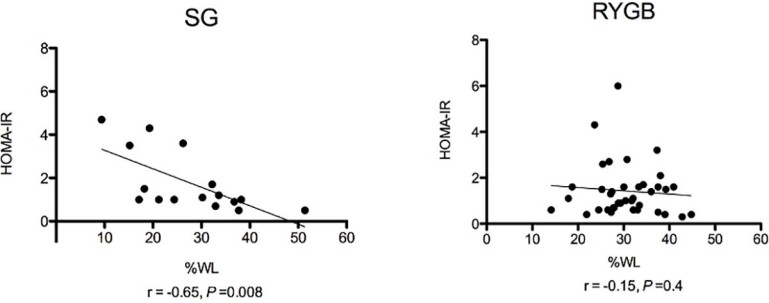

**Fig 4. HOMA-IR and correlation with percentage Weight Loss (%WL) after SG and RYGB.**

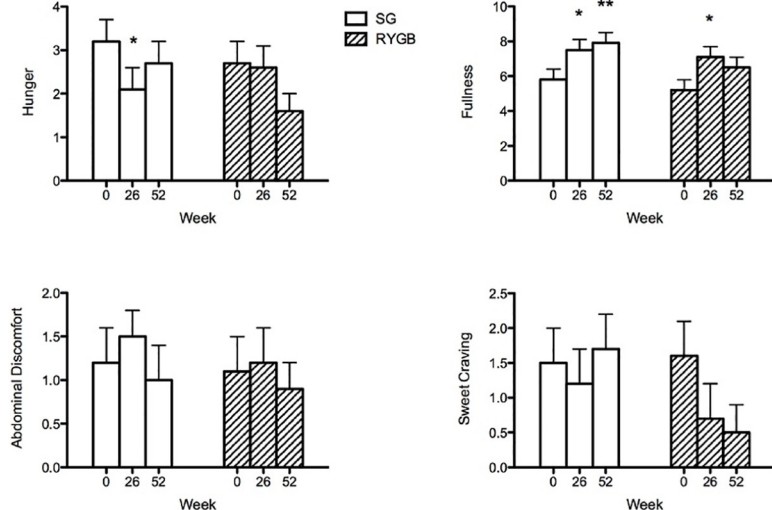

**Fig 5. Visual analog scale results of appetitive ratings.** Values are AUC 0–120 min. *$P < 0.05$, **$P < 0.01$, ***$P < 0.001$ within group difference.

remained a significant negative correlation between HOMA-IR and percent weight loss in SG (r = -0.59, $P = 0.04$) and no correlation in RYGB (r = -0.25, $P = 0.19$).

## Visual analog scale analysis

VAS ratings for appetitive sensations were similar in both groups at baseline. There were decreases in hunger after SG at 26 weeks ($P = 0.03$) that were not maintained at 1 year, whereas the opposite trend was observed after RYGB with a decrease in hunger at 1 year ($P = 0.09$). Both groups showed increases in fullness. There were no differences in abdominal discomfort. When asked "How much do you crave something sweet right now?" there was a trend towards decreased sweet cravings in RYGB compared to baseline ($P = 0.07$) and compared with SG ($P = 0.056$) at 52 weeks (Fig 5), whereas there were no significant changes in sweet cravings after SG.

## Discussion

This study demonstrates significant changes after bariatric surgery in levels of gut hormones that regulate glucose and energy homeostasis and shows differences in these levels between RYGB and SG. Increases in postprandial PYY and GLP-1 levels in the blood were observed primarily after RYGB and transiently after SG. Anatomic changes after RYGB that circumvent the gastric pylorus accelerates nutrient delivery to the distal intestine and likely stimulates secretion of these L cell hormones. Chronically high gastric emptying rates after RYGB have also been shown to induce adaptive changes such as an increase in enteroendocrine cell number, villous height, and surface area [33–35]. In SG there is increased gastric pressure, lack of receptive relaxation in the tubular stomach remnant, accelerated gastric emptying and small bowel transit to induce early and prolonged secretion of L-cell hormones from the distal intestine [36–39]. RYGB has been shown to induce higher and sustained postprandial PYY and GLP-1 compared to SG up to 1 year follow up [40–43]. We observed increases in PYY and GLP-1 in SG at 26 weeks which were not maintained at 52 weeks. Sleeve dilatation after SG

does occur in a majority of patients at one year follow up [44]. We postulate this could slow gastric emptying and subsequent GI hormone secretion though this has not been studied.

Changes in GLP-1 and PYY may underlie improvements in glucose homeostasis. Increases in postprandial insulin levels were seen in both groups, as early as 15 minutes following consumption of the test meal. Early postprandial peaks in insulin secretion, proportional to glucose, have been observed in patients with Type 2 diabetes undergoing RYGB, that correlated with improved hemoglobin A1C [45]. In addition to rapid nutrient transport and an early glycemic stimulus, GLP-1 contributes significantly to insulin secretion [46]. GLP-1 receptor blockade after RYGB significantly reduces insulin secretion and this reduction is greater than that seen in non-surgical controls [47]. The increase in PYY may also contribute to improved insulin sensitivity [21, 22].

Interestingly, HOMA-IR correlated with percentage weight loss only after SG, suggesting that improvement in insulin resistance is mediated, at least in part, by the amount of weight loss. In contrast, there was not a significant correlation between HOMA-IR and weight loss after RYGB. Thus, it is possible that weight loss independent changes, such as the increases in GLP-1 and PYY, are driving much of the improvement in HOMA-IR, with weight loss playing a somewhat lesser role. In conjunction with greater weight loss, additional gut hormones and changes in other factors not measured in this study such as metabolites, bile acids and the microbiome may promote improvement in glucose homeostasis which most studies show is greater after RYGB compared with SG in patients with type 2 diabetes mellitus [19, 43].

Both procedures result in a dysregulation of ghrelin when compared with non-surgical weight loss or laparoscopic adjustable gastric banding (LAGB), which result in a rise in fasting ghrelin and maintenance of meal-related decreases [25, 26]. In contrast, there was no significant change in ghrelin levels after RYGB, although it should be noted that by two years and later after RYGB, ghrelin levels do increase [26] and there was a large decrease after SG. In our study, pre-operative ghrelin levels were greater in the SG group compared with RYGB for unclear reasons. There was a wide range of fasting ghrelin values (SG, 226–960 pg/ml; RYGB, 110–904 pg/ml); this inter-individual variability and the smaller sample size in the SG group likely underlie the difference in baseline values. We cannot rule out the possibility that some of the variability in levels was due to the assay used which measures both acyl-ghrelin, considered the active form of ghrelin, plus des-acyl ghrelin. Other characteristics associated with differences in ghrelin levels, such as BMI, insulin resistance, and sex [48, 49] were similar between groups. Racial differences in ghrelin have been observed: African Americans had higher fasting levels and impaired postprandial suppression[50–52]. In our study, there was a larger proportion of African American subjects in the SG group compared to RYGB (36.8% and 15.0%, respectively), but there was no difference in fasting ghrelin between African American and Caucasian SG subjects ($P = 0.56$). Despite differences in ghrelin at baseline, the statistical analysis of change in ghrelin with time took into account baseline levels. Additionally, the percent reduction in AUC persisted at 52 weeks after SG. The large decrease in ghrelin is most likely due to resection of a large portion of the ghrelin-secreting cells of the gastric fundus and is consistent with the findings of recent studies [41, 42].

The mean fasting leptin level was also lower in RYGB compared to SG despite similar BMI at baseline. Body composition analysis was not performed thus, we are unable to decipher if differences in leptin were due to differences in adipose tissue mass or distribution. Menopause is known to be associated with decreased leptin and there was a higher proportion of women above the age of 50 in the RYGB group compared to SG (38% and 20%, respectively) which may have contributed somewhat to lower leptin levels. Given that the relationship between leptin and ghrelin is complex, [53] we are unable to draw inferences about whether differences in leptin levels between groups contributed to differences in ghrelin.

Paradoxically, decreases in post-prandial hunger were not maintained at 52 weeks despite marked decreases in ghrelin after SG. In addition, fullness increased significantly after SG despite lower postprandial PYY and GLP-1 levels compared to RYGB. In a study of 12 subjects who underwent SG, there were similar VAS scores for hunger and satiety at 3 months follow up compared to baseline despite increases in postprandial PYY [54]. Other mediators of fullness and satiety related to the mechanical changes after SG, such as increases in gastric pressure and vagal firing, could contribute to these findings. Interestingly, there was a trend toward decreased sweet cravings after RYGB, which recapitulates the findings of a prior study comparing LAGB with RYGB where LAGB patients had no change in sweet cravings and RYGB patients had significantly lower fasting and postprandial sweet cravings [26]. GLP-1 and PYY are potential mediators of this effect as they have been shown to enhance the ability to taste sweet flavors and heighten aversion to sweet taste [55, 56]. The presence of oral sweet taste receptors in the intestinal tract are known to enhance the intake and preference for sugar-rich foods and mediate GLP-1 release [57]. Our finding of increases in GLP-1 and PYY associated with RYGB support this mechanism and highlight the potential role of the gut-brain axis in mediating sugar preference recently described in rodents [58].

There was a significantly higher mean percentage weight loss after RYGB at 52 weeks. In line with long-term data, these differences in weight loss are anticipated to become more pronounced over time [11]. A subset of subjects was followed out to two years with weight data available for 74% of SG (14/19) and 70% of RYGB (28/40) demonstrating a lack of continued weight loss in SG and continued weight loss in RYGB of 26% and 32%, respectively.

A limitation of this study is the lack of randomization. There are also other gastrointestinal hormones of energy and glucose homeostasis of interest which were not measured [59, 60]. Additionally, only one type of meal stimulus was used. A solid meal stimulus or a different macronutrient composition, administered at different times of the day or at different rates, could have elicited different effects on hormone secretion and appetitive sensation. Rates of feeding have the ability to evoke higher sensations of satiety without corresponding changes in GLP-1 or PYY, which makes it possible other mechanisms unrelated to gut hormone secretion might underlie the success of surgery [54].

In conclusion, our results demonstrate that differences in gastrointestinal hormones of energy homeostasis and changes in sweet cravings are possible mechanisms for greater weight loss after RYGB compared to SG. Additionally, changes in these hormones may mediate weight-independent improvement in insulin resistance after RYGB. Continued research may further identify different factors that contribute to the metabolic benefits of SG and RYGB and the differences between these procedures.

## Supporting information

**S1 File. Dataset.**
(XLSX)

## Acknowledgments

We would like to thank Irene Conwell, Princess Swan and Mya Pugh for their expert technical assistance.

## Author Contributions

**Conceptualization:** Judith Korner.

**Data curation:** Rachel Arakawa, Gerardo Febres, Judith Korner.

**Formal analysis:** Bin Cheng, Judith Korner.

**Funding acquisition:** Judith Korner.

**Investigation:** Rachel Arakawa, Gerardo Febres, Abraham Krikhely, Marc Bessler, Judith Korner.

**Methodology:** Judith Korner.

**Project administration:** Rachel Arakawa, Gerardo Febres, Judith Korner.

**Resources:** Judith Korner.

**Software:** Judith Korner.

**Supervision:** Judith Korner.

**Validation:** Judith Korner.

**Visualization:** Judith Korner.

**Writing – original draft:** Rachel Arakawa, Judith Korner.

**Writing – review & editing:** Rachel Arakawa, Judith Korner.

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
