## [Decision Letter · Decision Letter 0]

23 Apr 2020

PONE-D-20-08332

Prospective study of gut hormone and metabolic changes after laparoscopic sleeve gastrectomy and Roux-en-Y gastric bypass

PLOS ONE

Dear Dr. Arakawa,

Thank you for submitting your manuscript to PLOS ONE. After careful consideration, we feel that it has merit but does not fully meet PLOS ONE’s publication criteria as it currently stands. Therefore, we invite you to submit a revised version of the manuscript that addresses all the points raised during the review process.

Particularly, you will see that additional informations regarding several aspects of your study have been asked by several reviewers. In addition, you will note that several comments have been made regarding the statistical analysis performed, the way some data are presented, interpretated, and discussed, and some missing important references in the bibliography.

We would appreciate receiving your revised manuscript by Jun 07 2020 11:59PM. To enhance the reproducibility of your results, we recommend that if applicable you deposit your laboratory protocols in protocols.io, where a protocol can be assigned its own identifier (DOI) such that it can be cited independently in the future. For instructions see: http://journals.plos.org/plosone/s/submission-guidelines#loc-laboratory-protocols

We look forward to receiving your revised manuscript.

Kind regards,

François Blachier, PhD

Academic Editor

PLOS ONE

Journal Requirements:

At this time we ask that you include the following in your manuscript:

1. The patient recruitment date range

2. Additional information regarding the VAS survey used in the study, ensuring that you have provided sufficient details so that others could replicate the analyses. For instance, if you developed the survey as part of this study and it is not under a copyright more restrictive than CC-BY, please include a copy as Supporting Information.

3. Please provide a sample size and power calculation in the Methods, or discuss the reasons for not performing one before study initiation.

4. Thank you for including your competing interests statement; "Dr. Korner participates on advisory boards for Applied Biosciences, Digma Medical, GI Dynamics, and Esquagama Labs.  Dr. Bessler is a founder of Endobetes, has a patent application, and owns stock in it.  Dr. Krikhely is a consultant for Intuitive Surgical, CLG, and CSATs.  These disclosures do not constitute a competing interest.  This does not alter our adherence to PLOS ONE policies on sharing data and materials.  "

We note that you have a patent relating to material pertinent to this article. Please provide an amended statement of Competing Interests to declare this patent (with details including name and number), along with any other relevant declarations relating to employment, consultancy, patents, products in development or modified products etc. Please confirm that this does not alter your adherence to all PLOS ONE policies on sharing data and materials, as detailed online in our guide for authors http://journals.plos.org/plosone/s/competing-interests by including the following statement: "This does not alter our adherence to  PLOS ONE policies on sharing data and materials.” If there are restrictions on sharing of data and/or materials, please state these. Please note that we cannot proceed with consideration of your article until this information has been declared.

Reviewers' comments:

Reviewer's Responses to Questions

**Comments to the Author**

1. Is the manuscript technically sound, and do the data support the conclusions?

Reviewer #1: Yes

Reviewer #2: Yes

Reviewer #3: Yes

Reviewer #4: Yes

Reviewer #5: Yes

2. Has the statistical analysis been performed appropriately and rigorously? 

Reviewer #1: Yes

Reviewer #2: Yes

Reviewer #3: No

Reviewer #4: Yes

Reviewer #5: No

3. Have the authors made all data underlying the findings in their manuscript fully available?

Reviewer #1: Yes

Reviewer #2: Yes

Reviewer #3: Yes

Reviewer #4: Yes

Reviewer #5: Yes

4. Is the manuscript presented in an intelligible fashion and written in standard English?

Reviewer #1: Yes

Reviewer #2: Yes

Reviewer #3: Yes

Reviewer #4: Yes

Reviewer #5: Yes

5. Review Comments to the Author

Reviewer #1: This manuscript describes a prospective study comparing the clinical outcomes and gut hormone response after two different bariatric surgery procedures, SG and RYBG. This research addresses a very timely and important topic, since SG being a much simpler surgical procedure has become ever more appealing and the most widely performed bariatric procedure worldwide. Yet, there is a growing amount of evidence suggesting that SG has poorer outcomes compared to RYGB, also supported by the conclusion of the herein described work. In sum, the knowledge gained by having conducted this study builds on previously available data and can help to support future clinical decision makers and guidance's. Despite the scientific merit, as well as this being a well designed, adequately conducted and very well described study, there are a number of points that still need to be addressed by the authors:

1- A similar proportion of patients in both groups had type 2 diabetes diagnosed prior to the bariatric surgery procedure. Information on disease duration and anti-diabetic treatment prior and after surgery, should be provided in the manuscript. This could allow to evaluate how comparable were the groups at baseline. Likewise the authors should provide further details on the progression of T2D after the procedures.

2- The authors should provide more detailed information on the VAS scale used in the methods section, including references and whether this was previously validated.

3- Figures with VAS data are presented using different Y axis scales, which give a false impression by magnifying scant differences that the authors classify as a trend when in fact there is no statistically difference. Likewise, data need be presented in the results and discussed in a more objective way. The effect of RYGB on sweet cravings is currently overemphasized and is not suported by the data.

4- Baseline ghrelin levels were significantly higher in the group of subjects submitted to SG, without any obvious explanation being provided by the authors nor hypothesis being advanced. Was the type of surgical procedure performed decided on the basis of baseline ghrelin levels? Authors should expand the discussion on this topic, and particularly addressing how these differences could represent a limitation for data interpretation.

5- Baseline leptin levels were also significantly higher in the SG group, suggesting that patients despite having a similar BMI could have a different body composition. Authors should also elaborate on possible hypothesis for this difference, this is finding is even more surprising given the fact that patients with higher fat mass tend to have higher leptin levels and lower fasting ghrelin levels.

6- In the study design the authors mention that patients were evaluated 2, 12, 26 and 52 weeks after surgery, although data is only depicted for baseline, 26 and 52 weeks after surgery. How did weight loss progressed in the initial post-operative period, was it similar?

7- In Table 1, waist should be expressed in cm and not in inches to comply with the use of the metric system as BMI.

8- In line 95 the sentence "PYY increases insulin sensitivity..." lacks a reference at the end (38), already in the reference list and quoted in the discussion.

Reviewer #2: Aim of the present study was that to evaluate differences in the effect of two procedures of bariatric surgery, particularly laparoscopic sleeve gastrectomy (SG) and Roux-en-Y gastric bypass (RYGB), on some gut hormones, such as fasting and post-prandial levels of PYY, GLP-1 and ghrelin, and glucometabolic homeostasis. Circulating levels of leptin and some subjective parameters such as hunger, fullness, sweet craving and abdominal disconfirm were also evaluated. The Authors conclude that differences in gut hormones, mainly PYY and GLP-1, and in glucometabolic homeostasis can be invoked to explain the greater efficacy of RYGB compared to SG, particularly after 1 year.

The topic is of interest despite the increasing adoption of SG against obesity depends upon the lower incidence of intra- and post-surgical complications when compared to RYGB.

The Introduction appropriately describes the most relevant articles dealing with the research field. The results are adequately presented and discussed.

Comments

- GLP-1 is a peptide having a short half-life due to the proteolytic action of DPP-IV (dipeptidyl-peptidase-4). Did the Authors add a DPP-4 inhibitor in blood tubes?

- The Authors have not mentioned the analytical methods for glucose, insulin and leptin.

- It’s difficult to establish whether the post-RYGB or post-SG weight loss is a consequence of differences in gut hormones secretion or induces a different gut hormones secretion. The Referee suggests to statistically adjust any parameter, particularly PYY, GLP-1 and ghrelin, by the weight loss reached at each time point. In other words, the significant differences at week 52 and not at week 26 vs. basal values or between RYGB and SG might disappear after an adjustment for weight loss. This issue should be discussed.

- Is there a particular reason to measure post-prandial circulating levels of GLP-1 until to 60 min?

- In Statistical Analysis there is no description of the method used to calculate the correlations.

- Line 101: new paragraph with “The objective of this…”

- Lines 168-169: is there a sex-matching in the two groups?

- Lines 236-243: a full description of the method used to evaluate subjective parameters such sweet craving should be located in Materials&Methods and not in Results.

- Lines 273-278: The paragraph is not clear.

- The Authors should discuss the results of the study by Rigamonti et al. (Endocrine, 2017;55:113-123). In particular, anorexigenic gut hormones would play a a negligible role in SG-induced weight loss. In other words, other mechanisms not related to gut hormones secretion might underlie the success of bariatric surgery. This critical issue should be reported in the limitations of the work (final part of the Discussion).

Reviewer #3: The is a cohort study comparing RYGB vs SG and is very interesting with clear objectives set out. However the manuscript could benefit from a revision of how results are presented and expansion on the methods section.

In the statistical analysis section prior to details re fitting a mixed model authors should mention how descriptive summaries are summarized, e.g continuous variables means(SD) or median(IQR) if skewed and and for categorical variables, counts (%ages) presented etc. Also mention what tests were used to compare compare in terms of baseline characteristics. Additionally looks like correlation was also looked at, this should be mentioned.

Authors should state what covariance structure was fitted in the model. And mention if model assumptions were met.

Table 1 - should be the baseline characteristics by group.

Add a Table 2 to include descriptive of outcomes at all timepoints with changes. Also to authors should decide whether to present actual changes (e.g week 26 - baseline) (week52 - baseline), OR whether to present percentage change.

Currently Table 1 is labelled as baseline characteristics and changes over time...and then a foot note to indicate what was presented in the linear mixed effect. This is very confusing as the reader is left to work out changes etc...

Better to then have a Table 3 - which will include estimates from the model for the RYGB vs SG.

Also have full names in the table header e.g sleeve gastrectomy not SG.

For all these time-points measured, authors do not mention any issues with missing data and if at all how it was handled? Which would helpful to include 'n' available at each time points for the descriptive of outcomes presented in Table 1.

Reviewer #4: The paper entitled “Prospective study of gut hormone and metabolic changes after laparoscopic sleeve gastrectomy and Roux-en-Y gastric bypass” by Arakawa et al. aims at comparing and contrasting gut hormones levels (especially PYY Ghrelin and GLP-1) during a meal test in subjects operated on RYGB and SG. A quality of this study is that it is a prospective cohort studying meal test response at base line and 26 and 52 weeks post-surgery.

Although the study is well designed and the paper well written, it presents an important limitation. The authors report a statistical difference in ghrelin and leptin levels at baseline. Baseline is supposed to be before the surgery and the authors cannot compare the impact of surgery on patients’ gut hormone levels with such a difference at baseline.

This raise questions on the drastic reduction of ghrelin they observed at weeks 26 and 52 in subjects operated on SG. Is it because SG ghrelin levels are over evaluated at baseline or are the RYGB ghrelin levels under evaluated at baseline ? In both cases the interpretations could be different and RYGB could also result in a reduction of ghrelin levels although at a lesser extent that SG but in agreement with previous reports.

How can the authors explain such a difference? Are the 2 groups of subjects coming from the same center ? Were the meal tests performed and blood samples drawn within the same center ? Were the assays performed in the same center ? Can one imagine a problem of inhibitors during one of the blood drawn ? Or a problem of sample conservation before the ghrelin assay ?

In addition, the GLP-1 level profiles are also very different between the VSG group and the RYGB group at baseline (especially T15, T90 and T120 on figure 1) with no explanation.

Minor point

Line 165 the sentence “There were 59 subjects enrolled consisting of 63% Hispanic, 37% non-Hispanic, 22% African American, and 78% Caucasian adults” is confusing as the total is more than 100%

Are the AUC incremental or total AUC ?

Why are the AUC calculated for GLP1 different from the others ?

Reviewer #5: This article presents some interesting data related to the beneficial effect of bariatric surgery regarding weight loss and glucose tolerance by comparing the effects of vertical sleeve gastrectomy (VSG) and Roux-en-Y gastric bypass (RYGB). In this study, 19 patients undergoing VSG and 40 RYGB have been followed at different time points, before 6 months and one year after surgery, without randomisation between the two groups. At each time point, the patients were given a liquid meal and some gut hormones where measured at different time points. Overall, the study is correctly designed and provides interesting information but is not very new as several articles have already focused on the effects of these surgery on gut hormones, either independently or comparing both procedures. Generally speaking, this paper confirms the main literature showing a higher GLP-1 and PYY response after RYGB compared to VSG. The main interest resides in the time follow up suggesting a diminution of the VSG surgery effect on the long-time which is just discussed by a probable reversion of the VSG by stomach dilation. We can regret that the high individual variability is not discussed.

I have some major and minor comments, but I consider this research acceptable for publication if these points are correctly addressed

Major comments:

- The literature review about the comparison between VSG and RYGB and the role of gut hormones in these surgery is lacking the most recent published evidences (Svane et al, Gastroenterology 2019 ; Alamuddin et al, Obesity Surgery 2017 ; Casajoana et al, Obesity Surgery 2017 ; Cavin et al, Gastroenterology 2017; Larraufie et al, Cell Reports, 2019 …). The authors should discuss the most recent papers and how their results match or bring novelty to the existing litterature.

- The data analysis should be reviewed and better explained. I really appreciated data availability and this is very important for open-science, however I was quite surprised to see many missing points that are not discussed in the paper regarding gut hormone measurements. As an example, 3 individuals do not have any value for PYY measurement in the VSG group at 52 weeks and 6 in the RYGB and many data is missing for 15 minutes measurement despite being often the most important point regarding gut hormones. It would be important that the authors indicate clearly the number of measurement for each point on their graphs or tables. Moreover, the statical analysis is inappropriate as the authors use a linear regression with mixed effects to account for individual measurements for comparison between groups and time points, as well as to infer values for missing points in table one. This results in calculated values that differs to the actual mean using the raw data. As an example, the indicated mean value for PYY in the table in the VSG group at 52 weeks is 75 pM whereas the mean from the provided data is 85.4 pM! The use of a linear mixed model is wrong here due to the missing values which are not randomly distributed but associated with the variables individuals and time points. Moreover, due to the high heterogeneity of the response between individuals and through time, they cannot fit with linear models. I therefore encourage the authors to revisit their analysis.

- Data analysis suggest important individual variability in some of the different parameters analysis. It would be interesting to further analyse individual correlation between gut hormone levels and metabolic parameters as some papers have suggested that gut hormones can be predictive of bariatric surgery outputs, and this data could help confirming or not this.

Minor comments:

- The authors focus only on Ghrelin, PYY and GLP-1 as gut hormones without mentioning the potential role of other gut hormones such as GIP, CCK or Neurotensin in the beneficial effects of bariatric surgery.

- In the methods, there is no indication of blood sampling at 15 minutes despite data being presented. Moreover, it is not clear how time is considered: is the time indicated from the start or of the end liquid meal and did the authors measured time consumption during each meal, and did the authors notice any difference ? This information is quite important as they allowed a large amount of time to drink the meal which represents a large volume for patients after these surgery, and may therefore importantly alter the gut hormone response happens within minutes in response to food intake, especially in these patients.

- Total Ghrelin is measured, but nothing is said about the importance nor the origin of acylated ghrelin which is mainly produced in the stomach whereas non-acylated ghrelin is found in the upper SI. Can the authors discuss about this difference ?

- As often, RYGB procedure varies from one individual to the other and is dependent on the surgeon appreciation. An important parameter seems to be the length of SI that is bypass and that can explain the stimulation of more or less distant enteroendocrine cells. Does the authors have access to this information, and did they notice any correlation between the length of bypassed SI and responses?

Small remark:

Figure 1 PYY SG time point indicates 5minutes, I suppose this should be 15.

Overall, this study provides additional data showing that VSG and RYGB differently modulate some gut hormones that could explain the different physiological effects of these surgeries, mainly on the long term as VSG effects seem to decrease. However, the data analysis should be reviewed based on two points: the analysis presented here using linear models with mixed effects is not suitable due to the missing values that should not be inferred, especially considering the high intra and inter individual variability responses and an analysis individual centred could help to better characterize the high variability of responses.

I therefore consider this study worth publishing, but it requires the data analysis to be revisited to ensure appropriate tools have been used and a better acknowledgement of the existing literature.

6. PLOS authors have the option to publish the peer review history of their article (what does this mean?). If published, this will include your full peer review and any attached files.

Reviewer #1: No

Reviewer #2: No

Reviewer #3: No

Reviewer #4: No

Reviewer #5: Yes: Pierre Larraufie

---

## [Author Response · Author response to Decision Letter 0]

18 Jun 2020

Dear Dr. Blachier, 

We would like to thank you for the opportunity to submit a revised version of our manuscript entitled “Prospective study of gut hormone and metabolic changes after laparoscopic sleeve gastrectomy and Roux-en-Y gastric bypass.” We appreciate the careful review and comments from the Reviewers. Below we have fully addressed each of the comments. 

RESPONSE to the EDITOR:

From the editor, to include in manuscript: 

1. Style requirements, including those for file naming. Follow: 

b. https://journals.plos.org/plosone/s/file?id=7797/Title%20Page%20-%20ONE%20Formatting.pdf

The Vancouver style template was edited in Endnote to have brackets around citations rather than parentheses. On the authorship page, we deemed it not necessary to add the equal author contribution denotation. File names for figures and supplemental information files have been updated per PLOS style requirements. 

2. Patient recruitment date range: 

The first subject was enrolled on 4/02/03 and the last subject was enrolled on 9/15/17. This information is now included in the manuscript.

3. VAS survey additional info (sufficient details ensuring others could replicate the analysis)

Subjects completed a validated VAS questionnaire. The reproducibility and availability to predict subsequent food intake has been shown in previous studies (Flint et al 2000, Stubbs et al 2000). The VAS questionnaire consists of 100-mm lines with words anchored at each end describing extreme sensations of hunger, satiety, sweet cravings, and nausea or abdominal discomfort. Subjects were asked to make a vertical mark across the line corresponding to their feelings. Quantification was performed by measuring the distance from the left end of the line to the mark. This information has been added to the manuscript.

4. Sample size and power calculation in the methods (or discuss reasons for not performing one before study initiation)

A sample size and power calculation was not performed before study initiation, however, sample number was based on prior and ongoing work from our group comparing RYGB and laparoscopic adjustable gastric banding (LAGB) in which differences in gut hormone levels were observed in cross-sectional and prospective studies of subjects after RYGB (n= 9-28) and LAGB (n= 9-18) (references below). In the current study there are 40 and 19 subjects in the RYGB and SG groups, respectively. This information has been added to the manuscript.

Korner J, Inabnet W, Febres G, Conwell IM, McMahon DJ, Salas R, et al. Prospective study of gut hormone and metabolic changes after adjustable gastric banding and Roux-en-Y gastric bypass. Int J Obes (Lond). 2009;33(7):786-95.

Korner J, Inabnet W, Conwell IM, Taveras C, Daud A, Olivero-Rivera L, et al. Differential effects of gastric bypass and banding on circulating gut hormone and leptin levels. Obesity (Silver Spring). 2006;14(9):1553-61.

Korner J, Bessler M, Inabnet W, Taveras C, Holst JJ. Exaggerated glucagon-like peptide-1 and blunted glucose-dependent insulinotropic peptide secretion are associated with Roux-en-Y gastric bypass but not adjustable gastric banding. Surg Obes Relat Dis. 2007;3(6):597-601.

5. We note that you have a patent relating to material pertinent to this article. Please provide an amended statement of Competing Interests to declare this patent (with details including name and number), along with any other relevant declarations relating to employment, consultancy, patents, products in development or modified products etc. Please confirm that this does not alter your adherence to all PLOS ONE policies on sharing data and materials, as detailed online in our guide for authors http://journals.plos.org/plosone/s/competing-interests by including the following statement: "This does not alter our adherence to PLOS ONE policies on sharing data and materials.” If there are restrictions on sharing of data and/or materials, please state these. Please note that we cannot proceed with consideration of your article until this information has been declared.

Dr. Korner participates on advisory boards for Digma Medical, GI Dynamics, and Esquagama Labs. Dr. Bessler is a founder of Endobetes, has a patent application titled “Lumen Reinforcement and Anchoring System” application no. 16/046592. The patent is focused on a method to anchor devices in the GI tract and is assigned to Endobetes, a medical device startup with a goal of developing endoscopic devices for treatment of obesity and diabetes. Dr. Bessler owns stock in Endobetes. Dr. Krikhely is a consultant for Intuitive Surgical, CLG, and CSATs. These disclosures do not constitute a competing interest. This does not alter our adherence to PLOS ONE policies on sharing data and materials.

The Competing Interests statement was amended in the cover letter to include more details on the patent held by Dr. Bessler. Dr. Korner’s participation on the advisory board for Applied Biosciences was also removed.

The dataset was added to the Supporting Information section at the end of the manuscript and captioned “S1 file. Dataset”. The file name was changed to “S1_file.xlsx”

RESPONSE to REVIEWERS

Reviewer #1: This manuscript describes a prospective study comparing the clinical outcomes and gut hormone response after two different bariatric surgery procedures, SG and RYBG. This research addresses a very timely and important topic, since SG being a much simpler surgical procedure has become ever more appealing and the most widely performed bariatric procedure worldwide. Yet, there is a growing amount of evidence suggesting that SG has poorer outcomes compared to RYGB, also supported by the conclusion of the herein described work. In sum, the knowledge gained by having conducted this study builds on previously available data and can help to support future clinical decision makers and guidance's. Despite the scientific merit, as well as this being a well designed, adequately conducted and very well described study, there are a number of points that still need to be addressed by the authors:

1- A similar proportion of patients in both groups had type 2 diabetes diagnosed prior to the bariatric surgery procedure. Information on disease duration and anti-diabetic treatment prior and after surgery, should be provided in the manuscript. This could allow to evaluate how comparable were the groups at baseline. Likewise the authors should provide further details on the progression of T2D after the procedures.

We do not have detailed information on medication use or diabetes duration. Based on fasting glucose measurements, none of our subjects progressed to diabetes. Given results of several published randomized trials and observational studies, much longer follow-up would be required to examine progression from the non-diabetic to diabetic state. 

The Reviewer raises an interesting point about whether diabetes status could have a modifier effect. We believe that one of the most intriguing findings from this study is the correlation of HOMA-IR and percentage weight loss, which is only significant in the SG group. This data was re-analyzed for just non-diabetics in order to remove possible confounding from the use of diabetes medications. The results remain essentially the same (RYGB r= -0.25, P =0.19; SG r= -0.59, P =0.04) This data has been added to the Results in the manuscript.

2- The authors should provide more detailed information on the VAS scale used in the methods section, including references and whether this was previously validated.

Subjects completed a validated VAS questionnaire. The reproducibility and availability to predict subsequent food intake has been shown in previous studies (Flint et al 2000, Stubbs et al 2000). Subjects completed a VAS questionnaire consisting of 100-mm lines with words anchored at each end describing extreme sensations of hunger, satiety, sweet cravings, and nausea or abdominal discomfort. Subjects were asked to make a vertical mark across the line corresponding to their feelings. Quantification was performed by measuring the distance from the left end of the line to the mark. This information has been added to the manuscript.

Flint A, Raben A, Blundell JE, Astrup A. Reproducibility, power and validity of visual analogue scales in assessment of appetite sensations in single test meal studies. Int J Obes Relat Metab Disord. 2000;24(1):38-48.

Stubbs RJ, Hughes DA, Johnstone AM, Rowley E, Reid C, Elia M, et al. The use of visual analogue scales to assess motivation to eat in human subjects: a review of their reliability and validity with an evaluation of new hand-held computerized systems for temporal tracking of appetite ratings. Br J Nutr. 2000;84(4):405-15.

3- Figures with VAS data are presented using different Y axis scales, which give a false impression by magnifying scant differences that the authors classify as a trend when in fact there is no statistically difference. Likewise, data need be presented in the results and discussed in a more objective way. The effect of RYGB on sweet cravings is currently overemphasized and is not supported by the data.

We certainly agree that using different Y axes when comparing RYGB to SG would be unacceptable, however, that is not how we presented the data. We are comparing appetitive sensations over time and between surgical groups as opposed to comparing sensations to each other (eg. hunger to fullness or hunger to sweet cravings); if the latter case was the objective, then we agree that the same Y axis would be appropriate. However, that is not the objective, and in fact, use of the same Y-axis amongst these different parameters would compress the bar graph to the point where the actual numbers will be difficult to see.

The intention was not to magnify scant differences between groups but to compare differences of the same variable over time and between groups. In order to do this, each variable has its own distinct range of values. 

The data for sweet cravings show about a 3 fold difference in the AUC for RYGB compared to SG (~1.7 vs 0.5) that while not statistically significant represents a trend as was previously published comparing RYGB and LAGB (Tsouristakis et al 2019). We recognize, and agree, that this is a “trend” and do not want to over-interpret that data, but do feel that there is potentially important biology worthy of further investigation, particularly given recent results in rodents demonstrating that the gut-brain axis mediates sugar preference (Tan et al 2020). We have addressed your concerns regarding overemphasis of this trend in the manuscript and have shortened the Discussion.

Tsouristakis AI, Febres G, McMahon DJ, Tchang B, Conwell IM, Tsang AJ, et al. Long-Term Modulation of Appetitive Hormones and Sweet Cravings After Adjustable Gastric Banding and Roux-en-Y Gastric Bypass. Obes Surg. 2019;29(11):3698-705.

Tan HE, Sisti AC, Jin H, Vignovich M, Villavicencio M, Tsang KS, et al. The gut-brain axis mediates sugar preference. Nature. 2020;580(7804):511-6.

4- Baseline ghrelin levels were significantly higher in the group of subjects submitted to SG, without any obvious explanation being provided by the authors nor hypothesis being advanced. Was the type of surgical procedure performed decided on the basis of baseline ghrelin levels? Authors should expand the discussion on this topic, and particularly addressing how these differences could represent a limitation for data interpretation.

It is unclear why baseline ghrelin was significantly higher in SG compared to RYGB. The choice of procedure was based on patient and surgeon preference and not on baseline ghrelin levels. The RYGB and SG samples were run simultaneously within the same assay. We also use a pool of plasma that is included in every assay to ensure there is no assay drift over time. Data has been reanalyzed and no outliers were identified. Baseline characteristics associated with differences in ghrelin, such as BMI and insulin resistance, were similar in both groups. Sex-related differences in ghrelin levels have been observed (Soriano-Guillen et al 2016, Sakao et al 2019) however both surgical groups had a similar percentage of female subjects 78% for SG and 80% for RYGB. Racial differences in fasting ghrelin have been observed, where obese African Americans had higher fasting ghrelin (Araneta 2012) and impaired postprandial ghrelin suppression (Brownley 2004, Fluitt 2013) compared to Caucasians. In our study there was a larger proportion of African American subjects in the SG group compared to RYGB (36.8% and 15% respectively). However, there were no differences in fasting ghrelin between black and Caucasian SG subjects (P=0.56) 

There was a wide range of fasting ghrelin values (SG 226-960 pg/ml; RYGB 110-904 pg/ml). This variability has been observed by many others and is not unique to our study. As RYGB had more than twice the number of subjects as SG, the difference in fasting ghrelin could simply reflect the smaller sample size. Most importantly, analysis of the change in ghrelin with time took into account the baseline ghrelin values. The reduction in ghrelin AUC persisted at 52 weeks after SG. This makes physiologic sense, as the ghrelin-secreting cells of the gastric fundus are excised during the surgery, and not explained solely by a higher AUC at baseline. In line with our findings, recent studies comparing RYGB and SG also show a significant suppression of fasting and postprandial ghrelin after SG (Svane et al 2019, Alamuddin et al 2016, Casajoana et al 2017). As requested, we have expanded the discussion on this topic in the manuscript. 

Soriano-Guillen L, Ortega L, Navarro P, Riestra P, Gavela-Perez T, Garces C. Sex-related differences in the association of ghrelin levels with obesity in adolescents. Clin Chem Lab Med. 2016;54(8):1371-6. 

Sakao Y, Ohashi N, Sugimoto M, Ichikawa H, Sahara S, Tsuji T, et al. Gender Differences in Plasma Ghrelin Levels in Hemodialysis Patients. Ther Apher Dial. 2019;23(1):65-72.

Araneta MR, Barrett-Connor E. Adiponectin and ghrelin levels and body size in normoglycemic Filipino, African-American, and white women. Obesity (Silver Spring). 2007;15(10):2454-62.

Brownley KA, Light KC, Grewen KM, Bragdon EE, Hinderliter AL, West SG. Postprandial ghrelin is elevated in black compared with white women. J Clin Endocrinol Metab. 2004;89(9):4457-63.

Fluitt MB, Gambhir KK, Nunlee-Bland G, Odonkor W. Fasting plasma ghrelin levels are reduced, but not suppressed during OGTT in obese African American adolescents. Ethn Dis. 2013;23(4):436-40.

Svane MS, Bojsen-Moller KN, Martinussen C, Dirksen C, Madsen JL, Reitelseder S, et al. Postprandial Nutrient Handling and Gastrointestinal Hormone Secretion After Roux-en-Y Gastric Bypass vs Sleeve Gastrectomy. Gastroenterology. 2019;156(6):1627-41 e1.

Alamuddin N, Vetter ML, Ahima RS, Hesson L, Ritter S, Minnick A, et al. Changes in Fasting and Prandial Gut and Adiposity Hormones Following Vertical Sleeve Gastrectomy or Roux-en-Y-Gastric Bypass: an 18-Month Prospective Study. Obes Surg. 2017;27(6):1563-72.

Casajoana A, Pujol J, Garcia A, Elvira J, Virgili N, de Oca FJ, et al. Predictive Value of Gut Peptides in T2D Remission: Randomized Controlled Trial Comparing Metabolic Gastric Bypass, Sleeve Gastrectomy and Greater Curvature Plication. Obes Surg. 2017;27(9):2235-45.

5- Baseline leptin levels were also significantly higher in the SG group, suggesting that patients despite having a similar BMI could have a different body composition. Authors should also elaborate on possible hypothesis for this difference, this is finding is even more surprising given the fact that patients with higher fat mass tend to have higher leptin levels and lower fasting ghrelin levels.

Fasting leptin was lower in RYGB compared to SG despite similar BMI at baseline. Body composition analysis was not performed in this study thus we are unable to know whether differences in baseline leptin levels were due to differences in fat mass or adipose tissue distribution. If higher leptin levels were due to greater fat mass, we would also expect to see lower ghrelin levels in the SG group, which is not the case. Other factors known to affect leptin (eg. age, sex) were similar between groups. Menopause is known to be associated with decreased leptin and there was a higher proportion of women above the age of 50 in the RYGB group compared to SG (38% and 20% respectively) which may have contributed somewhat to lower leptin levels. We have added these issues to the Discussion including lack of body composition analysis as a study limitation. Given that the relationship between leptin and ghrelin is complex and somewhat controversial (Cummings and Foster 2003) we are unable to draw inferences about whether differences in leptin levels between groups contributed to differences in ghrelin.

Cummings DE, Foster KE. Ghrelin-leptin tango in body-weight regulation. Gastroenterology. 2003;124(5):1532-5.

6- In the study design the authors mention that patients were evaluated 2, 12, 26 and 52 weeks after surgery, although data is only depicted for baseline, 26 and 52 weeks after surgery. How did weight loss progressed in the initial post-operative period, was it similar?

 We have added a graph of the weight loss progression. (see Figure 1).All values are significant from baseline without between group differences until 52 weeks.

7- In Table 1, waist should be expressed in cm and not in inches to comply with the use of the metric system as BMI.

Waist circumference has been converted from inches to centimeters in the original table 1 (now labeled table 2).

8- In line 95 the sentence "PYY increases insulin sensitivity..." lacks a reference at the end (38), already in the reference list and quoted in the discussion.

Vrang et al 2006 was inserted as a reference for line 95

Reviewer #2: Aim of the present study was that to evaluate differences in the effect of two procedures of bariatric surgery, particularly laparoscopic sleeve gastrectomy (SG) and Roux-en-Y gastric bypass (RYGB), on some gut hormones, such as fasting and post-prandial levels of PYY, GLP-1 and ghrelin, and glucometabolic homeostasis. Circulating levels of leptin and some subjective parameters such as hunger, fullness, sweet craving and abdominal disconfirm were also evaluated. The Authors conclude that differences in gut hormones, mainly PYY and GLP-1, and in glucometabolic homeostasis can be invoked to explain the greater efficacy of RYGB compared to SG, particularly after 1 year.

The topic is of interest despite the increasing adoption of SG against obesity depends upon the lower incidence of intra- and post-surgical complications when compared to RYGB.

The Introduction appropriately describes the most relevant articles dealing with the research field. The results are adequately presented and discussed.

Comments

1- GLP-1 is a peptide having a short half-life due to the proteolytic action of DPP-IV (dipeptidyl-peptidase-4). Did the Authors add a DPP-4 inhibitor in blood tubes?

The Reviewer raises an important issue. Because of the susceptibility to proteolytic activity, we intentionally measured total GLP-1 which does not necessitate the use of a DPP-IV inhibitor. The assay used measures both the active form and inactive form when cleaved by DPP-IV. As much of GLP-1 is cleaved immediately upon secretion from the L-cell (Jens Holst, personal communication), and prior to the blood collected in the test tube, the measurement of active GLP-1 does not really assess how much is secreted from the L-cell as the much of the GLP-1 is already proteolytically processed and inactivated by the time it’s measured. For the Reviewer’s information, we have also measured total GLP-1 with and without addition of a DPP-IV inhibitor and have obtained similar results confirming that the presence of inhibitor is not necessary when measuring total peptide.

2- The Authors have not mentioned the analytical methods for glucose, insulin and leptin.

This information was referred to in an earlier manuscript (Korner et al 2005) and included below and now added to the manuscript. 

• Plasma glucose was measured by the hexokinase method

• Plasma insulin was measured with the Immulite Analyzer (Diagnostic Products Corp., Los Angeles, CA) with the lower limit of detection of 2uIU/ml.

• Leptin was measured with a human RIA kit (LINCO Research, Inc., St. Charles, MO) using a 125I-iodinated human leptin tracer.

• Total plasma immunoreactive ghrelin was measured by an RIA kit (Phoenix Pharmaceuticals, Belmont, CA) using 125I-iodinated ghrelin tracer and a rabbit polyclonal antibody against full-length, octanoylated human ghrelin that recognizes the acyl and des-acyl forms of the hormone, with the lower limit of detection of 20 pg/ml. 

3- It’s difficult to establish whether the post-RYGB or post-SG weight loss is a consequence of differences in gut hormones secretion or induces a different gut hormones secretion. The Referee suggests to statistically adjust any parameter, particularly PYY, GLP-1 and ghrelin, by the weight loss reached at each time point. In other words, the significant differences at week 52 and not at week 26 vs. basal values or between RYGB and SG might disappear after an adjustment for weight loss. This issue should be discussed.

Differences in gut hormones were seen at 26 weeks, where there was similar weight loss between the two procedures, so any differences in gut hormone secretion is likely due to the procedure itself and not due to the weight loss. At 52 weeks, the additional weight loss may produce some additional changes on the gut hormones; alternatively, the changes in gut hormones might influence weight loss. If one change is a consequence of the other, then adjusting for weight loss may erroneously negate the finding. We do agree with the Reviewer, that we cannot establish causation with this type of observational study. 

4- Is there a particular reason to measure post-prandial circulating levels of GLP-1 until to 60 min?

The RIA assay used to measure GLP-1 was quite labor intensive, involving use of iodinated tracer and extraction of samples. Since most of the postprandial increase in GLP-1 occurs within the first 60 min, and GLP-1 levels are nearly back to baseline by 60 min we sought to save technician time and expense without sacrificing important information. During the course of this study, a reliable ELISA did become commercially available, but we did not want to measure samples with two different assays over time.

5-In Statistical Analysis there is no description of the method used to calculate the correlations.

Pearson correlation was calculated. We have specified this in the methods.

6- Line 101: new paragraph with “The objective of this…”

This was changed in the manuscript

7- Lines 168-169: is there a sex-matching in the two groups?

The choice of bariatric procedure was based on the preference of the patient and surgeon and therefore sex matching was not performed in the two groups. For SG (n=19) there were 4 males (21%) and 15 females (79%). For RYGB (n=40) there were 8 males (20%) and 32 females (80%). This information is included in the manuscript.

8- Lines 236-243: a full description of the method used to evaluate subjective parameters such sweet craving should be located in Materials&Methods and not in Results.

The methods section was updated with a full description of the visual analog scale (VAS)

9-Lines 273-278: The paragraph is not clear.

We have changed the paragraph to read:

Interestingly, HOMA-IR correlated with percentage weight loss only after SG, suggesting that improvement in insulin resistance is mediated, at least in part, by the amount of weight loss. In contrast, there was not a significant correlation between HOMA-IR and weight loss after RYGB. Thus, it is possible that weight loss independent changes, such as the increases in GLP-1 and PYY, are driving much of the improvement in HOMA-IR, with weight loss playing a somewhat lesser role. In conjunction with greater weight loss, additional gut hormones and changes in other factors not measured in this study such as metabolites, bile acids and the microbiome may promote improvement in glucose homeostasis which most studies show is greater after RYGB compared with SG in patients with type 2 diabetes mellitus [19, 43].

10- The Authors should discuss the results of the study by Rigamonti et al. (Endocrine, 2017;55:113-123). In particular, anorexigenic gut hormones would play a a negligible role in SG-induced weight loss. In other words, other mechanisms not related to gut hormones secretion might underlie the success of bariatric surgery. This critical issue should be reported in the limitations of the work (final part of the Discussion).

Subjects were asked to consume the test meal over a 15 minute period. Different outcomes in VAS might be observed with differences in the rate of food delivery. This was added to the limitations section, along with the fact that many other GI hormones were not measured. 

Rigamonti AE, Bini S, Rocco MC, Giardini V, Massimini D, Crippa MG, et al. Post-prandial anorexigenic gut peptide, appetite and glucometabolic responses at different eating rates in obese patients undergoing laparoscopic sleeve gastrectomy. Endocrine. 2017;55(1):113-23.

Reviewer #3: The is a cohort study comparing RYGB vs SG and is very interesting with clear objectives set out. However the manuscript could benefit from a revision of how results are presented and expansion on the methods section.

In the statistical analysis section prior to details re fitting a mixed model authors should mention how descriptive summaries are summarized, e.g continuous variables means(SD) or median(IQR) if skewed and for categorical variables, counts (%ages) presented etc. Also mention what tests were used to compare in terms of baseline characteristics. Additionally looks like correlation was also looked at, this should be mentioned.

Descriptive summaries are summarized directly below tables 1 and 2. Tests for comparing baseline characteristics included the t-test for age and the Wald test for the others. Pearson correlation was used and now specified in the manuscript.

Authors should state what covariance structure was fitted in the model. And mention if model assumptions were met.

Random intercept effects were included to account for within-subject correlation, which was equivalent to a compound symmetric covariance structure. No other covariance structures were explored due to limited sample size.

Table 1 - should be the baseline characteristics by group.

Table 1 with baseline characteristics was added to the manuscript. 

Add a Table 2 to include descriptive of outcomes at all timepoints with changes. Also to authors should decide whether to present actual changes (e.g week 26 - baseline) (week52 - baseline), OR whether to present percentage change.

Actual change is presented. 

Currently Table 1 is labelled as baseline characteristics and changes over time...and then a foot note to indicate what was presented in the linear mixed effect. This is very confusing as the reader is left to work out changes etc...

Better to then have a Table 3 - which will include estimates from the model for the RYGB vs SG.

 Percent change in ghrelin AUC from baseline was calculated in light of differences in fasting levels and is now reported in the Results section. Percent change in AUC for the remaining hormones was calculated but not included in the table, in order to avoid confusion and minimize redundancy with the actual AUC data in the table. 

We have also included a separate Table for some of the baseline characteristics.

Also have full names in the table header e.g sleeve gastrectomy not SG.

Full names are now used in the tables 

For all these time-points measured, authors do not mention any issues with missing data and if at all how it was handled? Which would helpful to include 'n' available at each time points for the descriptive of outcomes presented in Table 1.

Please see response to reviewer 5, item 2

Reviewer #4: The paper entitled “Prospective study of gut hormone and metabolic changes after laparoscopic sleeve gastrectomy and Roux-en-Y gastric bypass” by Arakawa et al. aims at comparing and contrasting gut hormones levels (especially PYY Ghrelin and GLP-1) during a meal test in subjects operated on RYGB and SG. A quality of this study is that it is a prospective cohort studying meal test response at base line and 26 and 52 weeks post-surgery.

Although the study is well designed and the paper well written, it presents an important limitation. The authors report a statistical difference in ghrelin and leptin levels at baseline. Baseline is supposed to be before the surgery and the authors cannot compare the impact of surgery on patients’ gut hormone levels with such a difference at baseline.

1. This raise questions on the drastic reduction of ghrelin they observed at weeks 26 and 52 in subjects operated on SG. Is it because SG ghrelin levels are over evaluated at baseline or are the RYGB ghrelin levels under evaluated at baseline ? In both cases the interpretations could be different and RYGB could also result in a reduction of ghrelin levels although at a lesser extent that SG but in agreement with previous reports.

How can the authors explain such a difference? Are the 2 groups of subjects coming from the same center ? Were the meal tests performed and blood samples drawn within the same center ? Were the assays performed in the same center ? Can one imagine a problem of inhibitors during one of the blood drawn ? Or a problem of sample conservation before the ghrelin assay ?

In addition, the GLP-1 level profiles are also very different between the VSG group and the RYGB group at baseline (especially T15, T90 and T120 on figure 1) with no explanation.

The subjects were recruited from two hospital sites in Manhattan (CUIMC and Harlem Hospital). All of the surgeries were performed at the two sites by 3 different surgeons who reportedly use similar techniques, however, it certainly is possible that there is some variability. All the meal tests were performed, and blood samples drawn, by author GF and completed within the same center. Blood samples are kept on ice and spun in a refrigerated centrifuge at 4 degrees C. The GLP-1 assay was performed without addition of a DPP-IV inhibitor (please see response to Reviewer 2, item 1). Plasma samples containing EDTA were all stored at -80ºC and assayed in duplicate. A plasma pool is included in each assay in order to ensure that there is no assay drift over the course of this study. Repeated freeze-thaw cycles were avoided. The GLP-1 profile appears different because the AUC curves erroneously included the 120min timepoint. GLP-1 AUC was determined from 0-60 minutes. (see Response to Reviewer 2, item 4). The curves were corrected.

In order to account for baseline variability, the statistical analysis took into account the baseline value of the variable being tested. Given the known inter-individual variability in hormone levels, some authors choose to only present their data as incremental change over time and don’t show baseline values. We chose to show the actual values. We further analyzed the data using percent change in AUC from baseline. At 26 weeks percent change was 46.6% for SG and -11.3% for RYGB (P <0.01) and at 52 weeks percent change was 45.2% for SG and -13.7% for RYGB (P <0.01). Given the similar percent changes in SG and RYGB at 26 and 52 weeks, we have confidence the differences we observed were real and not explained solely by a higher AUC at baseline in SG. This has been added to the manuscript in the Results and Discussion. 

Given the care that was taken in the processing and assaying of the samples, and accounting for baseline values by the statistical analysis in addition to calculating percent change, we feel that the conclusions drawn from this report are valid. 

Minor point

2.Line 165 the sentence “There were 59 subjects enrolled consisting of 63% Hispanic, 37% non-Hispanic, 22% African American, and 78% Caucasian adults” is confusing as the total is more than 100%

Both race and ethnicity data are included. Ethnicity was characterized as Hispanic or non-Hispanic (which adds to 100%). Race was characterized as African American or Caucasian (which adds to 100%). There were no Asian, American Indian or Alaska Native, Native Hawaiian or Other Pacific Islander subjects in our study thus these categories are not included under ethnicity. The baseline characteristics of the participants are included in a new table, labeled “Table 1” for clarification.

3.Are the AUC incremental or total AUC ?

Total AUC was used. This has been clarified in the manuscript.

4.Why are the AUC calculated for GLP1 different from the others ?

Please see response to Reviewer 2, item 4.

Reviewer #5: This article presents some interesting data related to the beneficial effect of bariatric surgery regarding weight loss and glucose tolerance by comparing the effects of vertical sleeve gastrectomy (VSG) and Roux-en-Y gastric bypass (RYGB). In this study, 19 patients undergoing VSG and 40 RYGB have been followed at different time points, before 6 months and one year after surgery, without randomisation between the two groups. At each time point, the patients were given a liquid meal and some gut hormones where measured at different time points. Overall, the study is correctly designed and provides interesting information but is not very new as several articles have already focused on the effects of these surgery on gut hormones, either independently or comparing both procedures. Generally speaking, this paper confirms the main literature showing a higher GLP-1 and PYY response after RYGB compared to VSG. The main interest resides in the time follow up suggesting a diminution of the VSG surgery effect on the long-time which is just discussed by a probable reversion of the VSG by stomach dilation. We can regret that the high individual variability is not discussed.

I have some major and minor comments, but I consider this research acceptable for publication if these points are correctly addressed

Major comments:

-1. The literature review about the comparison between VSG and RYGB and the role of gut hormones in these surgery is lacking the most recent published evidences (Svane et al, Gastroenterology 2019 ; Alamuddin et al, Obesity Surgery 2017 ; Casajoana et al, Obesity Surgery 2017 ; Cavin et al, Gastroenterology 2017; Larraufie et al, Cell Reports, 2019 …). The authors should discuss the most recent papers and how their results match or bring novelty to the existing litterature.

These most recently published studies were added as citations in our manuscript. Our results largely match the findings of Svane et al 2019, Alamuddin et al 2016, and Casaioana et al 2017, where RYGB subjects had higher postprandial PYY and/or GLP-1 and SG subjects had marked ghrelin suppression. With the exception of Casaioana, they were also cohort studies with at least 1 year follow up. Our study has more participants than the other studies.

Cavin et al 2017 was also added to the discussion, as it provides further support for histologic changes following RYGB, namely hyperplasia and an increased number of incretin producing cells. 

Larraufie et al 2019, describes GLP-1 as the main driver of insulin secretion after RYGB, consistent with findings from Salehi et al. It is a possible mechanism for superior improvement in glucose homeostasis after RYGB which we have added to the discussion section of the manuscript, though we did not find significant differences in insulin AUC between surgical groups.

Svane MS, Bojsen-Moller KN, Martinussen C, Dirksen C, Madsen JL, Reitelseder S, et al. Postprandial Nutrient Handling and Gastrointestinal Hormone Secretion After Roux-en-Y Gastric Bypass vs Sleeve Gastrectomy. Gastroenterology. 2019;156(6):1627-41 e1.

Alamuddin N, Vetter ML, Ahima RS, Hesson L, Ritter S, Minnick A, et al. Changes in Fasting and Prandial Gut and Adiposity Hormones Following Vertical Sleeve Gastrectomy or Roux-en-Y-Gastric Bypass: an 18-Month Prospective Study. Obes Surg. 2017;27(6):1563-72.

Casajoana A, Pujol J, Garcia A, Elvira J, Virgili N, de Oca FJ, et al. Predictive Value of Gut Peptides in T2D Remission: Randomized Controlled Trial Comparing Metabolic Gastric Bypass, Sleeve Gastrectomy and Greater Curvature Plication. Obes Surg. 2017;27(9):2235-45.

Cavin JB, Couvelard A, Lebtahi R, Ducroc R, Arapis K, Voitellier E, et al. Differences in Alimentary Glucose Absorption and Intestinal Disposal of Blood Glucose After Roux-en-Y Gastric Bypass vs Sleeve Gastrectomy. Gastroenterology. 2016;150(2):454-64 e9.

Larraufie P, Roberts GP, McGavigan AK, Kay RG, Li J, Leiter A, et al. Important Role of the GLP-1 Axis for Glucose Homeostasis after Bariatric Surgery. Cell Rep. 2019;26(6):1399-408 e6.

Salehi M, Gastaldelli A, D'Alessio DA. Blockade of glucagon-like peptide 1 receptor corrects postprandial hypoglycemia after gastric bypass. Gastroenterology. 2014;146(3):669-80 e2.

-2.The data analysis should be reviewed and better explained. I really appreciated data availability and this is very important for open-science, however I was quite surprised to see many missing points that are not discussed in the paper regarding gut hormone measurements. As an example, 3 individuals do not have any value for PYY measurement in the VSG group at 52 weeks and 6 in the RYGB and many data is missing for 15 minutes measurement despite being often the most important point regarding gut hormones. It would be important that the authors indicate clearly the number of measurement for each point on their graphs or tables. Moreover, the statical analysis is inappropriate as the authors use a linear regression with mixed effects to account for individual measurements for comparison between groups and time points, as well as to infer values for missing points in table one. This results in calculated values that differs to the actual mean using the raw data. As an example, the indicated mean value for PYY in the table in the VSG group at 52 weeks is 75 pM whereas the mean from the provided data is 85.4 pM! The use of a linear mixed model is wrong here due to the missing values which are not randomly distributed but associated with the variables individuals and time points. Moreover, due to the high heterogeneity of the response between individuals and through time, they cannot fit with linear models. I therefore encourage the authors to revisit their analysis.

We greatly appreciate the reviewer’s comments. While we generally agree with the reviewer on model selection for such a longitudinal study, we do have reasons to choose the model used in the paper. Specifically, our choice of model was based on the following considerations. First, the model had to account for within subject correlation and between subject variations. We looked at nonlinearity by treating the week variable as categorical to explore nonlinear temporal trends. Had we a larger data set, we would have considered a growth curve (instead of a linear relationship) model with random effects on all the coefficients (instead of one only on the intercept). Unfortunately, such a sophisticated model would yield unstable results due to limited sample size. Based on our experience, we would need at least 100 subjects per group for such a task. As a compromise, we chose to use the simplest mixed effects model, that is, one included only intercept random effects. Another reason we choose a conditional model as we did was that the method would be appropriate in the presence of missing data if the missing at random (MAR) assumption was reasonable. This is because that our model, which is a likelihood based model, used all available data, that is, a subject with part of data missing would still be included in the analysis. We agree with the reviewer that there might be reason to suspect the MAR assumption, particularly for PYY. However, such a violation can hardly be “confirmed” in a small data. To sum up, our choice of model was a compromise of what needed to be done and what could be done. Per reviewer’s comment, we have added above discussions as part of the limitation of the paper. 

-3.Data analysis suggest important individual variability in some of the different parameters analysis. It would be interesting to further analyse individual correlation between gut hormone levels and metabolic parameters as some papers have suggested that gut hormones can be predictive of bariatric surgery outputs, and this data could help confirming or not this.

We agree completely. Unfortunately, the study was not designed to address this question and we don’t have other metabolic parameters. For example, lipids are not routinely measured in the surgical care of these patients and this was not in our protocol. As such, we were only able to address insulin resistance as measured by HOMA-IR and as the paper describes, while resistance improves in both procedures, the mechanism by which this is achieved may differ. The change in HOMA-IR did not correlate with changes in the hormones measured. Our ongoing research is delving further into additional potential mechanisms. 

Minor comments:

-4.The authors focus only on Ghrelin, PYY and GLP-1 as gut hormones without mentioning the potential role of other gut hormones such as GIP, CCK or Neurotensin in the beneficial effects of bariatric surgery.

We certainly agree with the Reviewer’s comment. A limitation of this study is that there were many other hormones or interest with a potential role but were not measured (Dimitriadis 2017 and Pucci & Batterham 2019). This was added to the limitations section of the manuscript.

Dimitriadis GK, Randeva MS, Miras AD. Potential Hormone Mechanisms of Bariatric Surgery. Curr Obes Rep. 2017;6(3):253-65.

Pucci A, Batterham RL. Mechanisms underlying the weight loss effects of RYGB and SG: similar, yet different. J Endocrinol Invest. 2019;42(2):117-28.

-5.In the methods, there is no indication of blood sampling at 15 minutes despite data being presented. Moreover, it is not clear how time is considered: is the time indicated from the start or of the end liquid meal and did the authors measured time consumption during each meal, and did the authors notice any difference ? This information is quite important as they allowed a large amount of time to drink the meal which represents a large volume for patients after these surgery, and may therefore importantly alter the gut hormone response happens within minutes in response to food intake, especially in these patients.

The test meal was consumed by subjects within a 15-minute period and blood samples were drawn thereafter, starting from the point at which they completed the drink. The methods section was updated with this clarification

-6.Total Ghrelin is measured, but nothing is said about the importance nor the origin of acylated ghrelin which is mainly produced in the stomach whereas non-acylated ghrelin is found in the upper SI. Can the authors discuss about this difference ?

We cannot rule out the possibility that some of the variability in ghrelin levels were due to the assay used which measures both acyl, considered the active form of ghrelin, plus des-acyl ghrelin. To circumvent this issue, some samples were assayed in a two-site sandwich assay specific for acyl-ghrelin (Liu et al 2008), but the assay was not sensitive enough to detect very low levels of acyl-ghrelin in some subjects with very high BMI or in many of the subjects after SG. We have added this to the discussion.

Liu J, Prudom CE, Nass R, Pezzoli SS, Oliveri MC, Johnson ML, et al. Novel ghrelin assays provide evidence for independent regulation of ghrelin acylation and secretion in healthy young men. J Clin Endocrinol Metab. 2008;93(5):1980-7.

-7.As often, RYGB procedure varies from one individual to the other and is dependent on the surgeon appreciation. An important parameter seems to be the length of SI that is bypass and that can explain the stimulation of more or less distant enteroendocrine cells. Does the authors have access to this information, and did they notice any correlation between the length of bypassed SI and responses?

Surgeons at the Columbia University Irving Medical Center and one surgeon at Harlem Medical Center, who did his training at Columbia, performed the procedures and adhered to a uniform surgical protocol for RYGB: division of the jejunum 50-100cm distal to the ligament of Treitz and anastomosing the afferent biliopancreatic limb of the jejunum 100-150cm distally. We don’t believe there was enough variability between surgeons and procedures performed on individual patients to determine a correlation between the length of bypassed SI and responses, although this is an interesting question, and may certainly affect response with longer limb bypass procedures which are not routinely performed at our institution.

8. Small remark:

Figure 1 PYY SG time point indicates 5minutes, I suppose this should be 15.

Yes, this was a typo and was corrected. Thank you!

9.Overall, this study provides additional data showing that VSG and RYGB differently modulate some gut hormones that could explain the different physiological effects of these surgeries, mainly on the long term as VSG effects seem to decrease. However, the data analysis should be reviewed based on two points: the analysis presented here using linear models with mixed effects is not suitable due to the missing values that should not be inferred, especially considering the high intra and inter individual variability responses and an analysis individual centred could help to better characterize the high variability of responses.

I therefore consider this study worth publishing, but it requires the data analysis to be revisited to ensure appropriate tools have been used and a better acknowledgement of the existing literature.

We believe that the statistical analysis used has been justified and the limitations that the Reviewer points out have been addressed above. We have addressed this now in the manuscript. Additionally, as the Reviewer recommends, we have more fully acknowledged the existing literature.

---

## [Decision Letter · Decision Letter 1]

30 Jun 2020

Prospective study of gut hormone and metabolic changes after laparoscopic sleeve gastrectomy and Roux-en-Y gastric bypass

PONE-D-20-08332R1

Dear Dr. Arakawa,

We’re pleased to inform you that your manuscript has been judged scientifically suitable for publication and will be formally accepted for publication once it meets all outstanding technical requirements.

Kind regards,

François Blachier, PhD

Academic Editor

PLOS ONE

Additional Editor Comments (optional):

Reviewers' comments:

Reviewer's Responses to Questions

**Comments to the Author**

1. If the authors have adequately addressed your comments raised in a previous round of review and you feel that this manuscript is now acceptable for publication, you may indicate that here to bypass the “Comments to the Author” section, enter your conflict of interest statement in the “Confidential to Editor” section, and submit your "Accept" recommendation.

Reviewer #1: All comments have been addressed

Reviewer #3: All comments have been addressed

2. Is the manuscript technically sound, and do the data support the conclusions?

Reviewer #1: Yes

Reviewer #3: Yes

3. Has the statistical analysis been performed appropriately and rigorously? 

Reviewer #1: Yes

Reviewer #3: Yes

4. Have the authors made all data underlying the findings in their manuscript fully available?

Reviewer #1: Yes

Reviewer #3: No

5. Is the manuscript presented in an intelligible fashion and written in standard English?

Reviewer #1: Yes

Reviewer #3: Yes

6. Review Comments to the Author

Reviewer #1: The authors have adequately addressed the comments I have raised and I feel that this manuscript is now acceptable for publication.

Reviewer #3: (No Response)

7. PLOS authors have the option to publish the peer review history of their article (what does this mean?). If published, this will include your full peer review and any attached files.

Reviewer #1: No

Reviewer #3: No

---

## [Editor Report · Acceptance letter]

7 Jul 2020

PONE-D-20-08332R1 

Prospective study of gut hormone and metabolic changes after laparoscopic sleeve gastrectomy and Roux-en-Y gastric bypass 

Dear Dr. Arakawa:

I'm pleased to inform you that your manuscript has been deemed suitable for publication in PLOS ONE. Congratulations! Your manuscript is now with our production department. 

Kind regards, 

on behalf of

Dr. François Blachier 

Academic Editor

PLOS ONE